# Impact of the COVID-19 Pandemic on Elderly Patients with Spinal Disorders

**DOI:** 10.3390/jcm11030602

**Published:** 2022-01-25

**Authors:** Hidetomi Terai, Shinji Takahashi, Koji Tamai, Yusuke Hori, Masayoshi Iwamae, Masatoshi Hoshino, Shoichiro Ohyama, Akito Yabu, Hiroaki Nakamura

**Affiliations:** 1Department of Orthopaedic Surgery, Osaka City University Graduate School of Medicine, Osaka 545-8585, Japan; hterai@med.osaka-cu.ac.jp (H.T.); koji.tamai.707@gmail.com (K.T.); yusukehori0702@gmail.com (Y.H.); yabuakito@gmail.com (A.Y.); hnakamura@med.osaka-cu.ac.jp (H.N.); 2Department of Orthopaedic Surgery, Shimada Hospital, Osaka 583-0875, Japan; iwamae0519@gmail.com; 3Department of Orthopaedic Surgery, Osaka City General Hospital, Osaka 534-0021, Japan; hoshino717@gmail.com; 4Department of Orthopaedic Surgery, Nishinomiya Watanabe Hospital, Nishinomiya 662-0863, Japan; a03ma012@yahoo.co.jp

**Keywords:** COVID-19, pandemic, spinal disorder, elderly, exercise habit, female, quality of life, activities of daily living

## Abstract

During the ongoing coronavirus disease 2019 (COVID-19) pandemic, home-quarantine has been necessary, resulting in lifestyle changes that might negatively affect patients with spinal disorders, including a reduction in their quality of life (QoL) and activities of daily living (ADLs). However, studies on this impact are lacking. This study aimed to investigate the age-related changes in QoL and ADLs in patients with spinal disorders, and also identify factors associated with decline in ADLs. This multicenter cross-sectional study included patients who visited four private spine clinics for any symptoms. The study participants either had a clinic reservation, were first-time clinic visitors, or had a return visit to the clinic. The participants completed several questionnaires at two points: pre-pandemic and post-second wave. Changes in patient symptoms, exercise habits, ADLs, and health-related QoL were assessed. A logistic regression model was used to calculate the odds ratio (OR) of each variable for decline in ADLs. We included 606 patients; among them, 281 and 325 patients were aged <65 and ≥65 years, respectively. Regarding exercise habits, 46% and 48% of the patients in the <65 and ≥65-year age groups, respectively, did not change their exercise habits. In contrast, 40% and 32% of the patients in the <65 and ≥65-year age groups, respectively, decreased their exercise habits. In the multivariate analysis, the adjusted ORs for sex (female), decreased exercise habit, and age >65 years were 1.7 (1.1–2.9), 2.4 (1.4–3.9), and 2.7 (1.6–4.4), respectively. In conclusion, there was a decline in the ADLs and QoL after the COVID-19 outbreak in patients with spinal disorders. Aging, reduction of exercise habits, and female sex were independent factors related to decline in ADLs.

## 1. Introduction

The coronavirus disease 2019 (COVID-19) pandemic has continued worldwide since early 2020 [1]. The pandemic was declared by the World Health Organization on 11 March 2020, with alarming levels of spread and severity [2]. In Japan, a state of emergency in response to the COVID-19 pandemic was declared between 7 April and 25 May 2020. People were required to stay at home during this period, with their lifestyles changing even after the end of the declaration. During this period, most gyms for elderly people were closed because the novel severe acute respiratory syndrome coronavirus 2 (SARS-CoV-2) is primarily transmitted through direct routes, including respiratory droplets and direct person-to-person contact [3,4]. Specifically, age ≥ 65 years is a known risk factor for severe acute respiratory infection by COVID-19 [5]. Consequently, people within that age category have more compromised activity compared with younger adults. Additionally, disuse in the elderly population increases their vulnerability to rapid skeletal muscle atrophy, functional strength loss, and multiple related negative health consequences [6].

The Japanese Orthopedic Association recommends the guidelines of the Centers for Medicare and Medicaid Services and American College of Surgeons [7,8], that is, that low acuity and intermediate acuity practice should be postponed or rescheduled. The North American Spine Society has provided guidelines for applying injections, interventional procedures, and surgeries, with the understanding that decision-making is strongly influenced by multiple factors, as follows [9]. The current pandemic might negatively affect patients with spinal disorders, including by reducing quality of life (QoL) and activities of daily living (ADLs), especially in the elderly population. However, there has been no study of this impact. Therefore, the present study aimed to reveal the changes in the QoL and ADLs of patients with spinal disorders after the first and second waves of the pandemic according to age, and to investigate the factors related to deterioration in ADLs.

## 2. Materials and Methods

### 2.1. Study Design

This was a multicenter cross-sectional study in four spine clinics. All patients who visited the spine clinics for any symptoms were asked to participate in a survey. Patients who provided informed consent were enrolled and were allowed to withdraw at any point if they wished (i.e., during or after completing the questionnaire). The participants were asked to answer questions from several questionnaires at two points: pre-pandemic and post-second wave. This survey was conducted between 1 November 2020 and 31 December 2020 in Osaka, Japan. In Japan, the first and second waves of the COVID-19 pandemic occurred in April and August, respectively, and the third wave occurred during the study period. 

All study participants provided written informed consent. The study protocol was approved by the Institutional Review Board of the representative institution (approval No: 2020-242).

### 2.2. Patients

Patients who had a reservation in the clinic, were visiting our spine clinic for the first time, or had a return visit to the clinic were enrolled in this study after providing written informed consent. Patients who could not understand the questionnaires were excluded from the analysis in order to ensure that the data obtained was accurate. Patients with new symptoms after the outbreak were excluded from the analysis investigating outcome changes.

Among 1103 patients who visited our spine clinics, 747 patients were enrolled in this study. Among those 747 patients with spinal disorders, 141 patients were excluded since their symptoms had developed after the COVID-19 outbreak. Therefore, we included 606 patients in the analysis. There were 281 and 325 patients in the <65- and ≥65-year groups, respectively (Table 1); the respective average ages were 49.0 (standard deviation, SD: 10.7) and 75.9 (6.3) years. The proportion of females was higher in the <65-year group (45%) than in the ≥65-year group (53%) (*p* = 0.039). There was no between-group difference in the ratio of patients who refrained from visiting clinics (37% and 32%, *p*-value = 0.228). Regarding exercise habits, 46% and 48% of patients in the <65- and ≥65-year groups, respectively, did not change their exercise habits. In contrast, 40% and 32% of patients in the <65- and ≥65-year groups, respectively, decreased their exercise habits. Compared with pre-pandemic symptoms, the current symptoms were worse in 8% and 7% of patients in the <65- and ≥65-year groups, respectively, with no significant between-group difference (*p*-value = 0.988).

### 2.3. Procedures for Data Collection

The study coordinator gave short instructions and the patients filled out the paper questionnaire forms with pencils. The paper questionnaires were returned on the day of the clinic visit. To deal with the risk of social-desirability bias, the questionnaires guaranteed anonymity and responses were compiled [10].

### 2.4. Instruments

The authors worked independently and then agreed on the final questionnaire design by proving feedback on content accuracy, wording, question order, and survey structure. Adjustments were progressively included by considering the feedback that emerged [11]. When full agreement among experts was achieved, the survey was started.

#### 2.4.1. General Information

The questions requested participants’ date of birth, sex, and whether they were reluctant (whether they had hesitated or not hesitated) to visit the hospital (Questionnaire items in Appendix A).

#### 2.4.2. Changes after the COVID-19 Pandemic

Changes after the COVID-19 pandemic were assessed with respect to changes in patient symptoms (improved, deteriorated, stable, or newly occurred after the pandemic), exercise habits (increased, decreased, stable, or no exercise habit), ADLs, and health-related QoL (HRQoL).

ADLs were evaluated using the criteria proposed by the long-term care insurance system of the Japanese Health and Welfare Ministry for evaluation of the degree of independence of disabled elderly individuals [12]. In rank J, despite the presence of disability, daily life is almost independent, and patients can leave the home without assistance from other individuals. In rank A, patients live independently indoors but require assistance to leave the home. In rank B, patients require some assistance living indoors and spend most of the day in bed, though they can sit up. Finally, in rank C, the patients spend all day in bed and require assistance with urination/defecation, dressing, and eating. We divided the ranks into two groups, that is, J, A (dependent or requires assistance to leave home) and B, C (bedridden or nearly bedridden).

HRQoL at both time points (pre-pandemic and post-second wave) was cross-sectionally assessed using the EuroQoL 5-dimension 5-level (EQ-5D) descriptive system in one survey. The EQ-5D measures HRQoL on a 1-–5 scale of five severity levels in five dimensions, including Mobility, Self-Care, Usual Activities, Pain/Discomfort, and Anxiety/Depression. The Japanese version of the EQ-5D-5L was used in this survey. Subsequently, the domain scores were converted into a summarized index based on previously published values [13].

### 2.5. Statistical Analysis

A restricted maximum-likelihood mixed-model regression was used to establish whether there was a significant difference in HRQoL between pre-pandemic and post-second wave. The model was used to assess the difference in HRQoL post-second wave between the <65- and ≥65-year groups. Additionally, we compared the number of patients who reached the minimum clinically important difference (MCID) between the <65- and ≥65-year groups. The index score required a change of at least 0.08 decline to reach the MCID [14] using a chi-square test. Each ADL at pre-pandemic and post-second wave was compared using Fisher’s exact test. Change in ADLs between pre-pandemic and post-second wave was compared using a chi-square test. Next, a binomial logistic regression model was used to calculate the odds ratio (OR) of each variable for ADLs decline (ADLs decline/no ADLs decline). The model was adjusted for potential confounding factors with a *p*-value < 0.05 in the univariate analysis, including age (<65/≥65 years), sex, regular exercise (decrease/no change after the outbreak), and ADLs status pre-pandemic. In the sensitivity analysis, the Mobility and Self-Care dimensions of the EQ-5D-5L were used to evaluate decline in ADLs. We defined decline in ADLs as one rank reduction of Mobility or Self-Care in EQ-5D-5L. Statistical significance was set at *p* < 0.05. Between-group comparisons of continuous and categorical variables were performed using t-tests and chi-square/Fisher’s exact tests, respectively. All *p*-values were two-sided. All statistical analyses were performed using SAS version 9.4 (SAS Institute Inc., Cary, NC, USA).

## 3. Results

### 3.1. Change in QoL and ADLs after the First and Second Waves of the Pandemic between the <65- and ≥65-Year Groups

Figure 1 shows the between-group comparison of the pre- and post-outbreak HRQoL measured using EQ-5D. The pre-outbreak EQ-5D scores in the <65- and ≥65-year groups were 0.89 (0.14) and 0.85 (0.17), respectively. The post-outbreak EQ-5D scores in the <65- and ≥65-year groups were 0.85 (0.15) and 0.79 (0.19), respectively. The mixed-effect model revealed a significant between-group difference, as well as between before and after the pandemic (both *p*-values < 0.001). There was no interaction between age (<65/≥65 years) and time (before and after the pandemic) (*p*-value = 0.139). The number of patients who reached the MCID were 47 (17%) vs. 51 (16%), respectively (*p*-value = 0.734).

Table 2 shows the between-group comparison of ADLs. Compared with the ≥65-year group, the <65-year group showed better pre-outbreak ADLs (*p* < 0.001). The pre-outbreak proportion of rank J1 was 77% and 58% in the <65- and ≥65-year groups, respectively. Moreover, the post-outbreak ADLs in the <65-year group was better than the pre-outbreak ADLs in the ≥65-year group (*p* < 0.001). However, the post-outbreak proportion of rank J1 decreased to 68% and 46% in the <65- and ≥65-year groups, respectively. Contrastingly, after the outbreak, the proportion of rank J2 increased from 21% to 29% and from 30% to 38% in the <65- and ≥65-year groups, respectively. There was greater deterioration in ADLs in the ≥65-year group than in the <65-year group (10% vs. 18%, *p*-value < 0.001).

### 3.2. Factors Related to Decline in ADLs

Table 3 shows the univariate and multivariate ORs for reduction in ADLs. The univariate analysis showed that for female patients, the OR of a reduction in ADLs was significantly increased in comparison to the male patients. The OR of a reduction in ADLs was also significantly increased in patients whose exercise habit had declined after the COVID-19 pandemic in comparison to the patients whose exercise habit had not changed or had improved. Additionally, the OR of a reduction in ADLs was significantly increased for the patients aged ≥65 years in comparison to the patients aged <65 years. There was no association between hesitation of visiting clinics and reduction in ADLs (*p*-value = 0.221). The pre-outbreak ADLs rank (per 1 rank increase) decreased the risk of reduction in ADLs. In the multivariate analysis, the adjusted ORs for sex (female), decrease of exercise habit, and age ≥ 65 years were 1.7 (1.1–2.9), 2.4 (1.4–3.9), and 2.7 (1.6–4.4), respectively. The adjusted OR for pre-outbreak ADLs rank (per 1 rank increase) was 0.3 (0.2–0.5).

Table 4 shows the univariate and multivariate ORs for reduction in ADLs using the Mobility or Self-care dimensions of EQ-5D-5L. There were 234 patients (39%) with reductions in ADLs. In the multivariate analysis, the adjusted ORs for decrease of exercise habit and age ≥65 years were 2.5 (1.5–4.1) and 1.9 (1.1–3.1), respectively.

## 4. Discussion

This is the first report to reveal the impact of the COVID-19 outbreak on status changes in patients with spinal disorders. This study sought to elucidate patients’ behavioral changes and the impact on their functional status during this pandemic. The patients’ HRQoL and ADLs were worse after the outbreak. There were physical and mental factors related to the decline in the HRQoL. Staying at home led to loss of opportunity to exercise and loneliness among individuals. This study reported a significant decrease in the EQ-5D by 0.04–0.06. Additionally, 16–17% of patients reached the MCID of EQ-5D. Upon the announcement of the first pandemic wave, leisure activities were closed for >1 month from March to May. Moreover, even after the end of the emergency declaration, lifestyles dramatically changed. Public health restrictions affect the physical activity of the elderly, especially those with pre-pandemic higher exercise/sports activity levels and lower HRQoL [15]. In the Chinese population, there were significant correlations among physical activity levels, HRQoL, and perceived stress levels [15]. Additionally, prolonged sitting time was found to negatively affect the HRQoL [15].

There was an obvious pandemic effect on ADLs, especially in elderly patients. This is consistent with a previous report detailing that muscle atrophy by disuse was more rapid and greater in elderly individuals than in young individuals [6]. This negative impact caused by the outbreak was more apparent in the elderly population. Additionally, they displayed a more significant decrease in exercise habits. During this pandemic period, there was a greater need for enforcing exercise programs for elderly people. Moreover, pandemic-related anxiety was found to be highest among citizens aged ≥65 years [16].

In our study, female sex was an independent risk factor for reduction in ADLs. Kim et al. [17] reported that disability in ADLs was more common in females (20.8% of the patients aged >65 years) than in males (13.3%). Moreover, compared with males, females showed a higher prevalence of chronic diseases, including arthritis, osteoporosis, and disc degeneration, which were risk factors for disability in ADLs. Furthermore, low back pain is more common in women [18]. Low physical activity might enforce back pain, which worsens chronic pain and results in low activity. In addition, the differences that exist between males and females in perception, expression, and pain tolerance are influenced by a wide variety of social and psychological factors [19]. Further, the incidence of knee osteoarthritis was much higher in females (71.9% of the patients aged 70–79 years) than in males (48.2%) [20]. Muscle weakness is a primary risk factor for pain, disability, and joint damage progression [21]. There might be sex differences in disuse muscle atrophy.

This study reported a positive relationship between exercise habits and decline in ADLs. Therefore, maintaining exercise habits is crucial for risk reduction. A recent systematic review highlighted that running may be a protective factor against the onset of low back pain based on studies investigating the incidence of low back pain in runners [22]. In this study, effective and safe remote rehabilitation was performed in 41.9% of patients with COVID-19, which facilitated rehabilitation in COVID-19-specialized general wards [23]. Additionally, telemedicine can provide very effective and satisfactory care in physical medicine and rehabilitative spine practice [24]. During the COVID-19 pandemic, there is a need for health services involving an integrated rehabilitation pathway to not only manage the numerous survivors, but also patients with spinal disease.

There was no association between hesitation in visiting clinics and decline in ADLs. Telemedicine might help to minimize risk of exposure for providers, in addition to allowing patients to stay at home and comply with public health recommendations during the pandemic, yet might have limited capability for thorough physical examination. However, Iyer et al. [25] proposed a simple remote examination method for use by spinal healthcare providers during telemedicine appointments to facilitate their ability to diagnose and treat patients. In Japan, spine surgeons performed interventions based on the prescription given at the last visit. Additionally, most patients who refrained from visiting might have had lower disease severity. Even during the emergency declaration, surgery was performed for emergent or urgent cases, including severe neurologic deficits, intractable pain, spinal trauma, and spinal infection.

This study has several limitations. First, recall bias should be considered since the questionnaire was completed after the outbreak [26]. Therefore, a simple question commonly used to determine ADLs in Japanese elderly individuals was used in order to prevent ambiguous answers. In addition, we confirmed the results of the sensitivity analysis using EQ-5D-5L Mobility and Self-Care dimensions. Second, in order to ensure that the questionnaire was easy for elderly participants to complete, it was designed such that it did not comprise detailed information. Therefore, we could not collect details regarding comorbidities and spine diseases, and did not collect information on treatment including medication, physical therapy, and surgery. This could be crucial for assessing patient status. Third, the sample size might be too small to analyze the association of ‘Hesitated to visit the clinic’ with reduction in ADLs. Fourth, this study involved only Japanese participants. Therefore, the results may not be generalizable to other populations. Finally, this study design could have led to a selection bias since the patients whose symptoms improved did not revisit the hospitals. This might have resulted in overestimates of reduction in ADLs and QoL. However, we believe that this did not affect the relationship between the factors and decline in ADLs.

## 5. Conclusions

This study revealed the decline in ADLs and QoL after the COVID-19 outbreak in patients with spinal disorders. Moreover, aging, reduction of exercise habits, and female sex were independent related factors for decline in ADLs. Therefore, there is an increased need for encouraging exercise for elderly people when the number of COVID-19 infections is remittent. In addition, we need safe and sustainable exercise programs for elderly people, even during the pandemic.

## Figures and Tables

**Figure 1 jcm-11-00602-f001:**
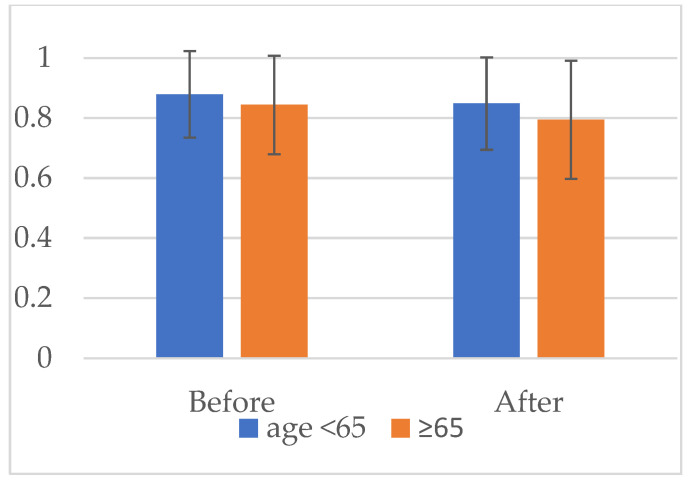
Comparison of pre- and post-pandemic health-related quality of life measured using the EQ-5D between patients aged <65 and ≥65 years. A mixed-effect model showed a significant difference between patients aged <65 and ≥65 years (*p*-value < 0.001), as well as before and after the pandemic (*p*-value < 0.001). There was no interaction between age (<65/≥65 years) and time (pre- and post-pandemic) (*p*-value = 0.139).

**Table 1 jcm-11-00602-t001:** Comparison of the patient characteristics between individuals aged <65 and ≥65 years.

	Age < 65 Years	Age ≥ 65 Years	*p*-Value
	*n* = 281	*n* = 325	
	*n* (%) or Mean (SD)	*n* (%) or Mean (SD)	
Age	49.0 (10.7)	75.9 (6.3)	<0.001
Sex (female)	126 (45)	173 (53)	0.039
Patients who refrained from visiting clinics due to the pandemic			
	103 (37)	104 (32)	0.228
Frequency of change of exercise habits before and after the outbreak			
No change	129 (46)	157 (48)	<0.001
More	17 (6)	3 (0.9)	
Less	89 (32)	130 (40)	
No exercise before or after the pandemic	46 (16)	35 (11)	
Comparison of the current and pre-outbreak symptoms			
Worse	19 (8)	20 (7)	0.988
Little worse	26 (11)	31 (11)	
No change	162 (71)	194 (71)	
Little better	11 (5)	15 (6)	
Better	9 (4)	12 (4)	

**Table 2 jcm-11-00602-t002:** Comparison of activities of daily living before and after the pandemic between patients aged <65 and ≥65 years.

	Pre-Pandemic			Post-Pandemic		
	Age < 65 Years	Age ≥ 65 Years	*p*-Value	Age < 65 Years	Age ≥ 65 Years	*p*-Value
	*n* (%)	*n* (%)		*n* (%)	*n* (%)	
ADLs rank						
Rank J1	215 (77)	189(58)	<0.001	192 (68)	149 (46)	<0.001
Rank J2	60 (21)	96 (30)		81 (29)	124 (38)	
Rank A1	0	26 (8)		1 (0.4)	30 (9)	
Rank A2	5 (2)	13 (4)		7 (2)	22 (7)	
Rank B	1 (0.4)	1 (0.3)		0	0	
ADLs change	<65	≥65				
	*n* (%)	*n* (%)				
Improved	5 (2)	9 (3)	<0.001			
No change	249 (89)	258 (79)				
Deteriorated	27 (10)	58 (18)				

Rank J1: Daily life is almost independent, and patients can go outside using different means of transportation without assistance from other individuals. Rank J2: Patients can go outside in the home vicinity without assistance from other individuals. Rank A1: Patients live independently indoors but require assistance to go out, and they stay out of bed for most of the day. Rank A2: Patients live independently indoors but require assistance to go out; however, they seldom go out and take several bed rests during the day. Rank B: Patients require some assistance living indoors and spend most of the day in bed; however, they can sit up.

**Table 3 jcm-11-00602-t003:** Factors associated with reduction in ADLs using the long-term care insurance system of the Japanese Health and Welfare Ministry.

	Univariate ORs (95% CI)	*p*-Value	Adjusted ORs (95% CI)	*p*-Value
Sex (female)	1.7 (1.1–2.8)	0.020	1.7 (1.1–2.9)	0.025
Hesitated to visit the clinic (yes)	1.3 (0.8–2.1)	0.221		
Decrease in exercise habits	2.2 (1.4–3.5)	0.001	2.4 (1.4–3.9)	<0.001
Age >65 years	2.0 (1.3–3.3)	0.004	2.7 (1.6–4.4)	<0.001
ADLs pre-pandemic (per 1 rank increase)	0.4 (0.2–0.6)	<0.001	0.3 (0.2–0.5)	<0.001

OR, odds ratio; CI, confidence interval; ADLs, activities of daily living.

**Table 4 jcm-11-00602-t004:** Factors associated with reduction in ADLs using the Mobility or Self-Care dimensions of EQ-5D-5L.

	Univariate ORs (95% CI)	*p*-Value	Adjusted ORs (95% CI)	*p*-Value
Sex (female)	1.3 (0.97–1.9)	0.079	1.5 (0.90–2.4)	0.124
Hesitated to visit the clinic (yes)	1.0 (0.7–1.5)	0.851		
Decrease in exercise habits	1.4 (0.98–1.9)	0.063	2.5 (1.5–4.1)	<0.001
Age > 65 years	1.5 (1.1–2.1)	0.015	1.9 (1.1–3.1)	0.016
Mobility dimension before pandemic (per 1 rank increase)	0.7 (0.5–0.92)	0.011	0.5 (0.3–0.7)	<0.001
Self-Care dimension before pandemic (per 1 rank increase)	0.7 (0.4–1.3)	0.315	1.4 (0.7–3.0)	0.294

OR, odds ratio; CI, confidence interval; ADLs, activities of daily living.

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
