# Peer review of "Impact of the COVID-19 Pandemic on Elderly Patients with Spinal Disorders"

_jcm, 2022, doi:10.3390/jcm11030602_

Round 1

Reviewer 1 Report

General appraisal

The authors conducted a multicenter cross-sectional study to investigate whether the COVID-19 pandemic affected HRQoL and ADL of patients that visited four spine clinics, regardless the condition with which they were coming to the clinic. Furthermore, the authors divided the patients in the sample into two groups: patients younger than 65 years old and patients older than 65 years old. They investigated whether the pandemic affected these two groups of patients differently. Different instruments were used to collect the data bevor the pandemic and after the first lockdown.

While the idea is interesting and the design the authors have chosen is appropriate, there are some aspects that have to be clarified before a decision can be made.

Suggestion to the Editor: Reject with the opportunity to resubmit after addressing the following issues.

Specific comments

Title:

Maybe the authors can be more specific with their title? Example: “Impact of the COVID-19 pandemic on HRQoL and ADL of elderly patients with spinal disorders – a cross-sectional study”

Introduction:

The aim of the study is not well defined! The autors stated “This study aimed (…)” but they investigated whether the COVID-19 pandemic induced HRQoL and ADL changes, and whether these changes were different in patients under and below 65 years old.

The authors are requested to reformulate the aims of the study. Maybe the authors should state different questions, which they would like to investigate within the study. For each question formulated in the introduction section of the manuscripts the authors have to provide under “Materials and Methods” – Statistical Analyses – the tool they have used to investigate it. For each question they have also to provide a result in the results section.

Material and Methods:

All patients who visited one of four private spine clinics were asked to participate. How many patients visited the clinics during the time period the study was running? How many patients did not give their consent?

The enrolled patients had:

  • A reservation in the clinic
  • Were visiting the clinic for the first time
  • Had a return visit to the clinic

What were the reasons for a return visit? This is an important issue! In case patients were returning to the clinic for a post-surgery visit, probably their HRQoL and ADL levels improved after surgery, even during the pandemic.

The authors should reorganize the “Materials and Methods” section. Start with a subtitle for “2.1. Study Design”.

Did the authors get ethic-committee approval to conduct this investigation? This information is missing in the entire manuscript and should be presented at the end of the paragraph of the subtitle “Study design”.

Add a second subtitle for “2.2. Patients”. Under this new subtitle the authors should give all information on inclusion vs. exclusion criteria.

How many patients visited the clinics? How many patients were asked to participate and refused? All this information is necessary to see whether attrition bias is present.

The subtitle “Questionnaire” should be reorganized. Maybe the authors can use the subtitle “Instruments” instead? They should state under this subtitle, which instruments they used.

Add a subtitle “Procedures for data collection”. The authors should describe, how data was collected. Paper and Pencil?

Statistical Analysis:

The authors state here that the “patients with new symptoms after the outbreak were excluded from the analysis (…)” This information should be provided under the subtitle “Patients”.

What was considered a new symptom? Why were patients with new symptoms excluded? Maybe the new symptoms appeared due to changes evoked by the pandemic?

The authors should reorganize the information provided under “statistical analysis”. For each question/aim stated in the introduction section, the authors should provide the statistical tool they have used to analyze it.

Mix-model regression detects within- and between-groups differences/changes. I suppose the model was used to test whether there were within group changes (time effect; before vs. after) and between-group effects (<65 vs. >65 y) for the measurement times pre- and post-pandemic. Right? The authors should clarify this.

Furthermore, a logistic regression model was used. It is not clear how the dependent variable (ADL decline) was calculated and coded? I suppose the dependent variable was binomial (ADL decline vs. no ADL decline). These information’s have to be provided.

Results:

The first sentence in the results section should be moved to the subtitle “Patients”: “Among 747 patients with spinal disorders, 141 patients were excluded since their 113 symptoms developed after the outbreak”. Why were the patients excluded when the symptoms developed during the outbreak?

The description and comparison of the patients in both groups should also be moved to the subtitle “Patients”. This is not a result, is a description of the sample.

Provide in this section a result for each aim/question raised in the introduction section.

When presenting the results, the authors should try to be as clear as possible.

Figure 1

In this figure the authors present the Health status index (EQ-5D-5L) of the patients in both groups before and after the outbreak. Maybe the authors can illustrate in the graph were significant difference were found. Are the significant differences found between the groups also clinically relevant? What is the MCID for the EQ-5D-5L health status index?

Table 1

“Comparison of the current and pre-outbreak symptoms” What kind of symptoms?

The authors used a specific ADL questionnaire, however one of the dimensions of the EQ-5D-5L assesses ADL activities and another dimension assesses mobility on a likert scale from 0 to 5. Did the authors thought about using this data for analysis? Independently from changes in Health Status Index, it would have been possible to see whether the levels of mobility decreased, remained or increased for each patient. The same for the dimension Activities of daily living.

Author Response

Dear Editor:

We would like to thank you for the careful review of our manuscript and for the insightful comments. We have revised our manuscript in accordance with your suggestions. Our point-by-point responses to the reviewers’ questions and comments are provided on the following pages, and the relevant revisions to the text have been made in blue font in the revised manuscript. Additionally, the revised manuscript underwent English language editing again for the correction of previous technical errors and to further improve grammar, readability, and consistency in phrasing.

We appreciate your consideration of our manuscript and believe that it has been significantly improved by your insightful suggestions. We look forward to working with you and the reviewers to move this manuscript closer to publication in JSM.

Sincerely,

Shinji Takahashi

Address: 1-4-3, Asahi-machi, Abeno-ku, Osaka 545-8585, Japan

Tel.: +81-06-6645-3851

Specific comments

Title:

Maybe the authors can be more specific with their title? Example: “Impact of the COVID-19 pandemic on HRQoL and ADL of elderly patients with spinal disorders – a cross-sectional study”

Introduction:

The aim of the study is not well defined! The autors stated “This study aimed (…)” but they investigated whether the COVID-19 pandemic induced HRQoL and ADL changes, and whether these changes were different in patients under and below 65 years old.

The authors are requested to reformulate the aims of the study. Maybe the authors should state different questions, which they would like to investigate within the study. For each question formulated in the introduction section of the manuscripts the authors have to provide under “Materials and Methods” – Statistical Analyses – the tool they have used to investigate it. For each question they have also to provide a result in the results section.

Response> We have revised the sentence for clarity. ‘Therefore, this study aimed to reveal the changes in the QoL and ADL of patients with spinal disorders after the first and second waves of the pandemic according to age and to investigate the factors related to ADL deterioration.’

Material and Methods:

All patients who visited one of four private spine clinics were asked to participate. How many patients visited the clinics during the time period the study was running? How many patients did not give their consent?

We enrolled all patients during the study period. However, patients who did not agree with this survey were excluded. We have added this information in the Study Design subsection.

‘Among 1103 patients who visited our spine clinics, 747 patients were enrolled in this study. ‘

The enrolled patients had:

  • A reservation in the clinic
  • Were visiting the clinic for the first time
  • Had a return visit to the clinic

What were the reasons for a return visit? This is an important issue! In case patients were returning to the clinic for a post-surgery visit, probably their HRQoL and ADL levels improved after surgery, even during the pandemic.

Response> We do not have any data on the reasons for a return visit, as we described in the section on limitations. Several patients might undergo surgery which might affect their QoL and ADL. In addition, we did not collect information on other treatments that patients might have undergone.

The authors should reorganize the “Materials and Methods” section. Start with a subtitle for “2.1. Study Design”.

Did the authors get ethic-committee approval to conduct this investigation? This information is missing in the entire manuscript and should be presented at the end of the paragraph of the subtitle “Study design”.

Response> Thank you for the comment. We have added this information in the Study Design subsection.

Add a second subtitle for “2.2. Patients”. Under this new subtitle the authors should give all information on inclusion vs. exclusion criteria.

How many patients visited the clinics? How many patients were asked to participate and refused? All this information is necessary to see whether attrition bias is present.

 Response> We do not have any information on the exact number of patients who visited the clinics. Most of the patients were asked to participate during the study period and none refused to participate in the study.

The subtitle “Questionnaire” should be reorganized. Maybe the authors can use the subtitle “Instruments” instead? They should state under this subtitle, which instruments they used.

 Response> Thank you for the comment. We have made this change accordingly.

Add a subtitle “Procedures for data collection”. The authors should describe, how data was collected. Paper and Pencil?

Response> Thank you for the comment. We have added the subtitle.

‘2.3 Procedures for data collection

The study coordinator gave short instructions and the patients filled out the paper questionnaire forms with pencils. The paper questionnaires were returned on the day of the clinic visit.’

Statistical Analysis:

The authors state here that the “patients with new symptoms after the outbreak were excluded from the analysis (…)” This information should be provided under the subtitle “Patients”.

What was considered a new symptom? Why were patients with new symptoms excluded? Maybe the new symptoms appeared due to changes evoked by the pandemic?

Response> We excluded patients with new symptoms because it was difficult to include them in the comparative analysis.

The authors should reorganize the information provided under “statistical analysis”. For each question/aim stated in the introduction section, the authors should provide the statistical tool they have used to analyze it.

Response> Thank you for the comment. We have reorganized the section as you suggested.

Mix-model regression detects within- and between-groups differences/changes. I suppose the model was used to test whether there were within group changes (time effect; before vs. after) and between-group effects (<65 vs. >65 y) for the measurement times pre- and post-pandemic. Right? The authors should clarify this.

Response> Thank you for the comment. As you mentioned, we used this model to detect the differences within (before and after) and between groups (HRQoL after pandemic between <65- and ≥65 years-groups) (fix effect = group, repeated measurement = time, random effect = patient ID).

We have revised these sentences accordingly.

Furthermore, a logistic regression model was used. It is not clear how the dependent variable (ADL decline) was calculated and coded? I suppose the dependent variable was binomial (ADL decline vs. no ADL decline). These information’s have to be provided.

 Response> Thank you for the comment. As you mentioned, we used ADL decline as binomial variable. We have revised this sentence accordingly.

Results:

The first sentence in the results section should be moved to the subtitle “Patients”: “Among 747 patients with spinal disorders, 141 patients were excluded since their 113 symptoms developed after the outbreak”. Why were the patients excluded when the symptoms developed during the outbreak?

Response> We excluded patients with new symptoms because there was no data for comparison such as the prevalence before pandemic.

The description and comparison of the patients in both groups should also be moved to the subtitle “Patients”. This is not a result, is a description of the sample.

 Response> Thank you for the comments. We have moved these sentences to the subtitle “Patients”.

Provide in this section a result for each aim/question raised in the introduction section.

When presenting the results, the authors should try to be as clear as possible.

  Response> Thank you for the comments. We have revised these sentences.

Figure 1

In this figure the authors present the Health status index (EQ-5D-5L) of the patients in both groups before and after the outbreak. Maybe the authors can illustrate in the graph were significant difference were found. Are the significant differences found between the groups also clinically relevant? What is the MCID for the EQ-5D-5L health status index?

   Response> Thank you for the comments. We analyzed using the MCID in the other field(0.08).

(AS Pickard, MP Neary, D Cella Estimation of minimally important differences in EQ-5D utility and VAS scores in cancer Health Qual Life Outcomes, 5 (2007) 70-7525-5-70)

Table 1

“Comparison of the current and pre-outbreak symptoms” What kind of symptoms?

Response> We do not have the detailed data about these symptoms.

The authors used a specific ADL questionnaire, however one of the dimensions of the EQ-5D-5L assesses ADL activities and another dimension assesses mobility on a likert scale from 0 to 5. Did the authors thought about using this data for analysis? Independently from changes in Health Status Index, it would have been possible to see whether the levels of mobility decreased, remained or increased for each patient. The same for the dimension Activities of daily living.

Response> This questionnaire for ADL evaluation is widely used in Japan to determine patients’ needs for long-term care insurance.

When we used Health Status Index, we defined ADL decline as one rank reduction of Mobility or Self-Care in EQ-5D-5L. There were 234 patients (39%) with ADL reduction.

Table 4. Factors associated with ADL reduction using Mobility or Self-Care of EQ-5D-5L

Univariate ORs

(95% CI)

P-value

Adjusted ORs

95% CI)

P-value

Sex (female)

1.3 (0.97-1.9)

0.079

1.5 (0.90-2.4)

0.124

Hesitated to visit the clinic (yes)

1.0 (0.7-1.5)

0.851

Decrease in exercise habits

1.4 (0.98-1.9)

0.063

2.5 (1.5-4.1)

<0.001

Age > 65 years

1.5 (1.1-2.1)

0.015

1.9 (1.1-3.1)

0.016

Mobility dimension  before pandemic (per 1 rank increase)

0.7 (0.5-0.92)

0.011

0.5 (0.3-0.7)

<0.001

Self-Care dimension  before pandemic (per 1 rank increase)

0.7 (0.4-1.3)

0.315

1.4 (0.7-3.0)

0.294

OR, odds ratio; CI, confidence interval; ADL, activities of daily living

Reviewer 2 Report

Dear authors

I realize that authors have many journals to consider when they want to publish their work, so I appreciate your interest in Journal of Clinical Medicine; I am very sorry not to be able to write in a more positive way. It is evident that you have put a great deal of effort into this project and I want to praise your efforts. Unfortunately, the actual contribution from your study is not clear and, the manuscript as currently written not suggests that it might be suitable for sharing information about these data, but the research that you reported, needs major edits. I should like to thank you for give me an opportunity to consider this work for publication. It may be that the you would like to consider resubmitting it, in which case I hope that the comments from my review may help you to revise it before resubmitting it. These comments are given below.

Best Regards

  • Introduction section: is too poor; to explain better the aims of the study;
  • Materials and Methods: this is the part that needs more revisions; enter approval number and institution of approval of the ethics committee or ethics committees as we are talking about four clinics; descrivere come è stato calcolato il numero del campione; come è stato deciso il numero minimo di soggetti da reclutare o che sarebbero stati necessari per definire lo studio fattibile; describe whether or not there were subjects previously affected by covid in the sample; you should provide additional information on how the patient - smoke, assessment of yellow flag or red flag, physical activity pre-pandemic period, work, etc; 

  • Discussion section: Discussions should be reviewed in light of the overall improvement of the paper. Redundant sentences and prewritten information should be avoided. Focus on take-home messages and how that information impacts the clinical practice
  • Limitations: expand this section as an isolated section by inserting many other limitations that the study has, such as that it refers to a small sample, Japanese-only population, etc.
  • Table 2: Insert a first row with description of the columns: before and after the pandemic    
  • Survey: insert in the appendix the questionnaire translated into English and in the original version, insert information on how it was created and if it was previously administered on a small sample to understand comprehensibility and applicability by the population under investigation;
  • Reference section: is poor. it is advisable to enter these references of questionnaires carried out with very detailed methods:

Bisconti M, Brindisino F, Maselli F. Gender Medicine and Physiotherapy: A Need for Education. Findings from an Italian National Survey. Healthcare (Basel). 2020 Nov 27;8(4):516. doi: 10.3390/healthcare8040516.

Viceconti A, Geri T, De Luca S, Maselli F, Rossettini G, Sulli A, Schenone A, Testa M. Neuropathic pain and symptoms of potential small-fiber neuropathy in fibromyalgic patients: A national on-line survey. Joint Bone Spine. 2021 Jul;88(4):105153. doi: 10.1016/j.jbspin.2021.105153

Maselli F, Esculier JF, Storari L, Mourad F, Rossettini G, Barbari V, Pennella D, Cataldi F, Viceconti A, Geri T, Testa M. Low back pain among Italian runners: A cross-sectional survey. Phys Ther Sport. 2021 Mar;48:136-145. doi: 10.1016/j.ptsp.2020.12.023

Rossettini G, Palese A, Geri T, Fiorio M, Colloca L, Testa M. Physical therapists' perspectives on using contextual factors in clinical practice: Findings from an Italian national survey. PLoS One. 2018 Nov 30;13(11):e0208159. doi: 10.1371/journal.pone.0208159.

Author Response

Dear Editor:

We would like to thank you for the careful review of our manuscript and for the insightful comments. We have revised our manuscript in accordance with your suggestions. Our point-by-point responses to the reviewers’ questions and comments are provided on the following pages, and the relevant revisions to the text have been made in blue font in the revised manuscript. Additionally, the revised manuscript underwent English language editing again for the correction of previous technical errors and to further improve grammar, readability, and consistency in phrasing.

We appreciate your consideration of our manuscript and believe that it has been significantly improved by your insightful suggestions. We look forward to working with you and the reviewers to move this manuscript closer to publication in JSM.

Sincerely,

Shinji Takahashi

Address: 1-4-3, Asahi-machi, Abeno-ku, Osaka 545-8585, Japan

Tel.: +81-06-6645-3851

  • Introduction section: is too poor; to explain better the aims of the study;

Response> We have changed the description of the aims for better clarity.

‘Therefore, this study aimed to reveal the changes in the QoL and ADL of patients with spinal disorders after the first and second waves of the pandemic according to age and to investigate the factors related to ADL deterioration.’

  • Materials and Methods: this is the part that needs more revisions; enter approval number and institution of approval of the ethics committee or ethics committees as we are talking about four clinics; descrivere come è stato calcolato il numero del campione; come è stato deciso il numero minimo di soggetti da reclutare o che sarebbero stati necessari per definire lo studio fattibile; describe whether or not there were subjects previously affected by covid in the sample; you should provide additional information on how the patient - smoke, assessment of yellow flag or red flag, physical activity pre-pandemic period, work, etc; 

Response> Thank you for the comments. We have added the sentences regarding ethical approval.

‘All study participants provided written informed consent. The study protocol was approved by the Institutional Review Board of the representative institution (approval No: 2020–242).’

We do not have any detailed information on the patients’ symptoms, smoking, or work. We have included the information on physical activity in ADL.

  • Discussion section: Discussions should be reviewed in light of the overall improvement of the paper. Redundant sentences and prewritten information should be avoided. Focus on take-home messages and how that information impacts the clinical practice

Response> Thank you for the comment. We have revised the Discussion section as you suggested.

  • Limitations: expand this section as an isolated section by inserting many other limitations that the study has, such as that it refers to a small sample, Japanese-only population, etc.

Response> Thank you for the comment. We have revised the Limitation section.

  • Table 2: Insert a first row with description of the columns: before and after the pandemic    

Response> Thank you for the comment. We have revised the table.

  • Survey: insert in the appendix the questionnaire translated into English and in the original version, insert information on how it was created and if it was previously administered on a small sample to understand comprehensibility and applicability by the population under investigation;

Response> Thank you for the comments. We have added the questionnaire in appendix.

  • Reference section: is poor. it is advisable to enter these references of questionnaires carried out with very detailed methods:

Response> Thank you for the comments. We have revised the references to include these.

Bisconti M, Brindisino F, Maselli F. Gender Medicine and Physiotherapy: A Need for Education. Findings from an Italian National Survey. Healthcare (Basel). 2020 Nov 27;8(4):516. doi: 10.3390/healthcare8040516.

Viceconti A, Geri T, De Luca S, Maselli F, Rossettini G, Sulli A, Schenone A, Testa M. Neuropathic pain and symptoms of potential small-fiber neuropathy in fibromyalgic patients: A national on-line survey. Joint Bone Spine. 2021 Jul;88(4):105153. doi: 10.1016/j.jbspin.2021.105153

Maselli F, Esculier JF, Storari L, Mourad F, Rossettini G, Barbari V, Pennella D, Cataldi F, Viceconti A, Geri T, Testa M. Low back pain among Italian runners: A cross-sectional survey. Phys Ther Sport. 2021 Mar;48:136-145. doi: 10.1016/j.ptsp.2020.12.023

Rossettini G, Palese A, Geri T, Fiorio M, Colloca L, Testa M. Physical therapists' perspectives on using contextual factors in clinical practice: Findings from an Italian national survey. PLoS One. 2018 Nov 30;13(11):e0208159. doi: 10.1371/journal.pone.0208159.

Round 2

Reviewer 1 Report

jcm-1516563-peer-review-v2

The authors addressed all questions and made significant improvements in the manuscript.

There are some little details that should be improved before the manuscript can be accepted for publication. Please see the pdf-file “jcm-1516563-peer-review v2_with_reviewer_comments” with my corrections and recommendations.

Author Response

Dear Editor and Reviewers,

We would like to thank you for the careful review of our manuscript and for the insightful comments. We have revised our manuscript in accordance with your suggestions. Our point-by-point responses to the reviewers’ questions and comments are provided on the following pages, and the relevant revisions to the text have been made in blue font in the revised manuscript. Additionally, the revised manuscript underwent English language editing again for the correction of previous technical errors and to further improve grammar, readability, and consistency in phrasing.

Line 152-153

Response> Thank you for the comment. We deleted the sentence.

Line 175-176: The authors should improve the way they report OR in the text. I think it would be better to report in this way: "The univariate analysis showed that female patients had significantly increased odds for a reduction in ADL in comparison to male patients."

Response> Thank you for the suggestion. We have changed the sentences.

Line225: When the authors report or discuss theirs results they should simple past. Example: "Staying hat home led to (...)"

Response> Thank you for the suggestion. We have changed the word.

Reviewer 2 Report

Dear

The your study needs few minor edits. I should like to thank you for give me an opportunity to consider this work for publication. It may be that the you would like to consider resubmitting it, in which case I hope that the comments from my review may help you to revise it before resubmitting it. These comments are given below. Best Regards

It is recommended to add all the following references in the Methods to support the questionnaire section:

Bisconti M, Brindisino F, Maselli F. Gender Medicine and Physiotherapy: A Need for Education. Findings from an Italian National Survey. Healthcare (Basel). 2020 Nov 27;8(4):516. doi: 10.3390/healthcare8040516.

Maselli F, Esculier JF, Storari L, Mourad F, Rossettini G, Barbari V, Pennella D, Cataldi F, Viceconti A, Geri T, Testa M. Low back pain among Italian runners: A cross-sectional survey. Phys Ther Sport. 2021 Mar;48:136-145. doi: 10.1016/j.ptsp.2020.12.023

Rossettini G, Palese A, Geri T, Fiorio M, Colloca L, Testa M. Physical therapists' perspectives on using contextual factors in clinical practice: Findings from an Italian national survey. PLoS One. 2018 Nov 30;13(11):e0208159. doi: 10.1371/journal.pone.0208159.

Author Response

Response> Thank you for the suggestion. We added the 3 papers in Reference.

Bisconti M, Brindisino F, Maselli F. Gender Medicine and Physiotherapy: A Need for Education. Findings from an Italian National Survey. Healthcare (Basel). 2020 Nov 27;8(4):516. doi: 10.3390/healthcare8040516.

Maselli F, Esculier JF, Storari L, Mourad F, Rossettini G, Barbari V, Pennella D, Cataldi F, Viceconti A, Geri T, Testa M. Low back pain among Italian runners: A cross-sectional survey. Phys Ther Sport. 2021 Mar;48:136-145. doi: 10.1016/j.ptsp.2020.12.023

Rossettini G, Palese A, Geri T, Fiorio M, Colloca L, Testa M. Physical therapists' perspectives on using contextual factors in clinical practice: Findings from an Italian national survey. PLoS One. 2018 Nov 30;13(11):e0208159. doi: 10.1371/journal.pone.0208159.